# ReasonNav: Human-Inspired Global Map Reasoning for Zero-Shot Embodied Navigation

## Abstract

Embodied agents often struggle with efficient navigation because they rely primarily on partial egocentric observations, which restrict global foresight and lead to inefficient exploration. In contrast, humans plan using maps: we reason globally first, then act locally. We introduce **ReasonNav**, a human-inspired framework that operationalizes this reason-then-act paradigm by coupling Multimodal Large Language Models (MLLMs) with deterministic planners. ReasonNav converts a top-down map into a discrete reasoning space by room segmentation and candidate target nodes sampling. An MLLM is then queried in a multi-stage process to identify the candidate most consistent with the instruction (object, image, or text goal), effectively leveraging the model's semantic reasoning ability while sidestepping its weakness in continuous coordinate prediction. The selected waypoint is grounded into executable trajectories using a deterministic action planner over an online-built occupancy map, while pretrained object detectors and segmenters ensure robust recognition at the goal. This yields a **unified zero-shot navigation framework** that requires no MLLM fine-tuning, circumvents the brittleness of RL-based policies and scales naturally with foundation model improvements. Across three navigation tasks, ReasonNav consistently outperforms prior methods that demand extensive training or heavy scene modeling, offering a scalable, interpretable, and globally grounded solution to embodied navigation.

## 1    Introduction

Embodied AI agents often face challenges in efficient navigation due to reliance on partial, egocentric observations, which limit global foresight leading to suboptimal, meandering trajectories. While existing methods incorporate global map information, these approaches are typically task-specific, require extensive training, or struggle to generalize across diverse goal types (Wen et al., 2024; Lin et al., 2025). In contrast, humans navigate by reasoning globally over maps before acting locally, enabling strategic planning even with coarse 2D floor plans and minimal real-time exploration. This contrast motivates a central question: *Can we endow embodied agents with human-inspired global map reasoning to enable zero-shot, goal-directed navigation across diverse tasks?*

Multimodal Large Language Models (MLLMs) appear promising for this challenge. When presented with a floor plan and an instruction such as "Bring the mug from the kitchen to the bedroom," current MLLMs can generate plausible high-level plans in a zero-shot manner. However, when tasked with embodied navigation, these models struggle. The reason lies in a fundamental mismatch: MLLMs are optimized for semantic reasoning, not for producing precise spatial coordinates or continuous control signals. They are excellent global reasoners but poor spatial controllers.

Our key insight is to embrace this mismatch by decomposing navigation into two complementary components. Instead of asking an MLLM to directly output coordinates, we transform navigation into a discrete reasoning problem. A set of well-distributed candidate is first generated using Poisson Disk Sampling over a top-down map. The MLLM then reasons in a multi-stage querying process to select the candidate that best aligns with the instruction, effectively leveraging its semantic reasoning capabilities while sidestepping its weakness in continuous spatial prediction. This global reasoning process is further strengthened by incorporating a model ensemble strategy to enhance robustness and accuracy. Once a target coordinate is selected, we ground this global plan into an executable trajectory using deterministic algorithms. Specifically, a hybrid A* + VFH* planner, operating on

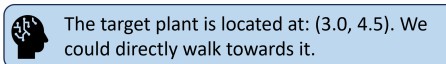
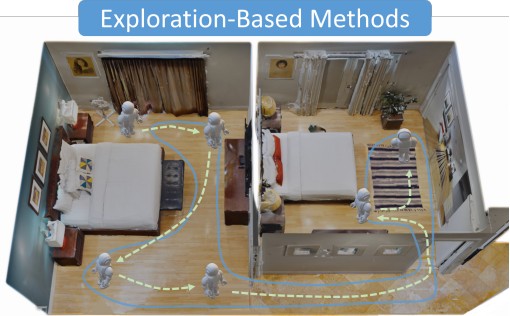
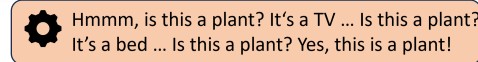

Figure 1: Main difference between our ReasonNav and previous exploration-based methods: in ReasonNav, after reasoning and obtaining the location of the desired target, the controlled agent will directly walk towards to object, whereas exploration-based methods heavily rely on extensive local semantic recognition or matching.

an online, wall-aware occupancy map, ensures reliable local path execution and collision avoidance. Robust recognition at the goal is achieved with pretrained detection and segmentation models.

This design yields several distinct advantages over prior work. By explicitly separating high-level reasoning from low-level control, ReasonNav implements a human-inspired *reason-then-act* paradigm that produces interpretable plans and avoids the inefficiency of reactive, exploration-heavy strategies, as shown in Figure 1. Because the framework relies purely on zero-shot reasoning without task-specific fine-tuning, ReasonNav unifies object-goal, image-goal, and text-goal navigation in a single framework, in contrast to fragmented approaches requiring separate models or training pipelines. Deterministic planners provide robustness and generalization, eliminating the instability, sample inefficiency, and sim-to-real challenges commonly faced by reinforcement learning methods. Finally, ReasonNav naturally benefits from ongoing improvements in foundation models: as MLLMs become stronger, their global reasoning quality directly enhances navigation performance, making the framework inherently scalable and future-proof. Readers are encouraged to view our supplementary video.

In summary, our contributions are as follows:

- We propose **ReasonNav**, a novel framework that integrates MLLM-based global reasoning with deterministic local planning, enabling a human-inspired *reason-then-act* paradigm for embodied navigation.
- ReasonNav provides a unified, zero-shot solution to diverse navigation tasks, including object-goal, image-goal, and text-goal navigation, without requiring task-specific fine-tuning or reinforcement learning.
- By reframing navigation as discrete global reasoning followed by robust grounding with A* + VFH*, ReasonNav achieves superior efficiency, reliability, and interpretability compared to prior approaches that rely on reactive exploration or complex scene modeling.

## 2 RELATED WORK

**Goal-Oriented Navigation.** Recent advances in embodied navigation can be broadly categorized into end-to-end learning-based methods and construction-based planning approaches. End-to-end methods, often based on reinforcement learning (RL), encode visual observations and directly predict low-level actions (Mousavian et al., 2019; Yang et al., 2018; Ye et al., 2021; Majumdar et al., 2023; Maksymets et al., 2021; Yadav et al., 2023; Sun et al., 2025; Chang et al., 2023). While effective for short-term action prediction, these approaches often struggle to capture long-horizon dependencies and efficiently memorize fine-grained context, limiting their ability to plan globally.

Construction-based planning methods address this limitation by building structured representations of the environment, such as top-down semantic maps (Zhang et al., 2024b; Lei et al., 2024; Kuang et al., 2024; Krantz et al., 2023), value maps (Yokoyama et al., 2023; Long et al., 2024), or 3D

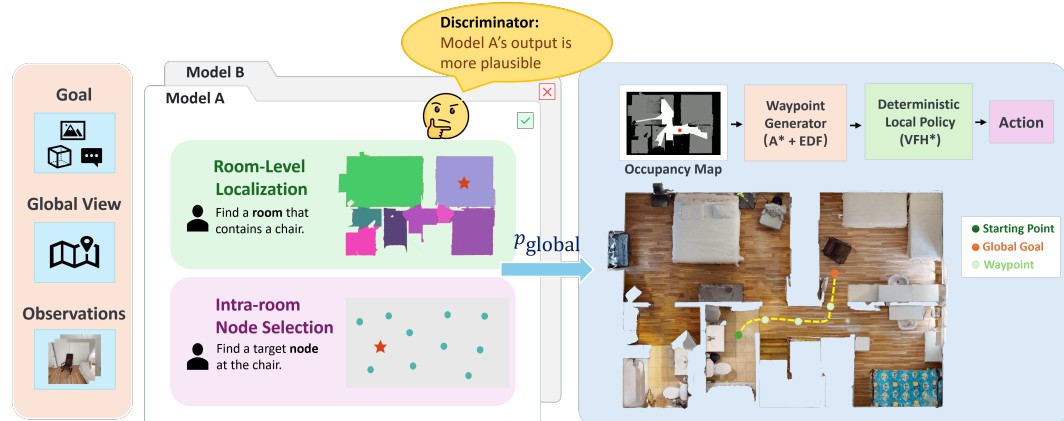

Figure 2: **The ReasonNav Framework** in two stages: 1) Global Reasoning, where a Multimodal Large Language Model (MLLM) reasons about a top-down map and the goal instruction through a multi-stage discrete selection process to determine a precise global target waypoint ($p_{\text{global}}$). This MLLM reasoning stage can be further enhanced by a model ensemble for increased robustness; 2) Local Navigation, where a deterministic planner safely guides the agent to the selected global waypoint using an online occupancy map.

scene graphs (Yin et al., 2024; 2025; Zhu et al., 2025). These methods update the map dynamically using online observations, enabling more informed path planning. However, because the map is constructed incrementally from local observations, global planning remains constrained, and agents may take sub-optimal paths or incur unnecessary exploration. Offline methods (Werby et al. (2024); Gu et al. (2023)) build dense, open-vocabulary 3D scene graphs prior to navigation, offering richer semantic priors but at the cost of significant computational overhead and long reconstruction times, which limits real-time applicability. Overall, while construction-based approaches improve over purely reactive RL agents, they either depend on dense pre-built representations or remain limited by incremental local observations, leaving room for more flexible and scalable global reasoning.

**Large Pretrained Models in Embodied Navigation.** The emergence of multimodal large language models (MLLMs) has introduced a new paradigm for navigation, leveraging their common-sense reasoning and generalization capabilities (Driess et al., 2023; Brohan et al., 2023; Chen et al., 2024; Dorbala et al., 2024; Yu et al., 2023). Following the Vision-Language-Action (VLA) paradigm (Brohan et al., 2023), methods such as Navid (Zhang et al., 2024a), NaviLLM (Zheng et al., 2024), and OctoNav (Gao et al., 2025) fine-tune MLLMs to directly map visual observations and instructions to low-level actions. Zero-shot approaches like NavGPT (Zhou et al., 2023a) and ESC (Zhou et al., 2023b) instead encode the agent's observation and action history as textual prompts to guide decision-making. Despite their reasoning power, invoking MLLMs at every timestep incurs high computational cost and latency, challenging real-time deployment.

**Limitations and Opportunities.** Taken together, prior work highlights a key trade-off: end-to-end RL methods offer reactive control but limited global foresight; construction-based methods improve planning but depend on incremental or dense scene representations; and MLLM-based methods excel at reasoning but are computationally expensive and often disconnected from robust low-level control. Our work, ReasonNav, addresses these gaps by combining the strengths of each paradigm: we leverage MLLM reasoning for global, zero-shot goal selection over a top-down map, while using deterministic planning for reliable local navigation. As shown in Figure 1, different from all the prior exploration-based methods, this integration enables interpretable, efficient, and scalable navigation without relying on dense scene reconstructions or fine-tuning the LLM, directly addressing the limitations of previous approaches.

## 3 METHOD

In this section, we present the architecture of our framework, as illustrated in Figure 2. We first define our versatile navigation tasks, then describe our hierarchical global reasoning module that identifies a target coordinate on a 2D map. We subsequently explain the local navigation policy for reaching this target and conclude with a model ensemble strategy for improved robustness.

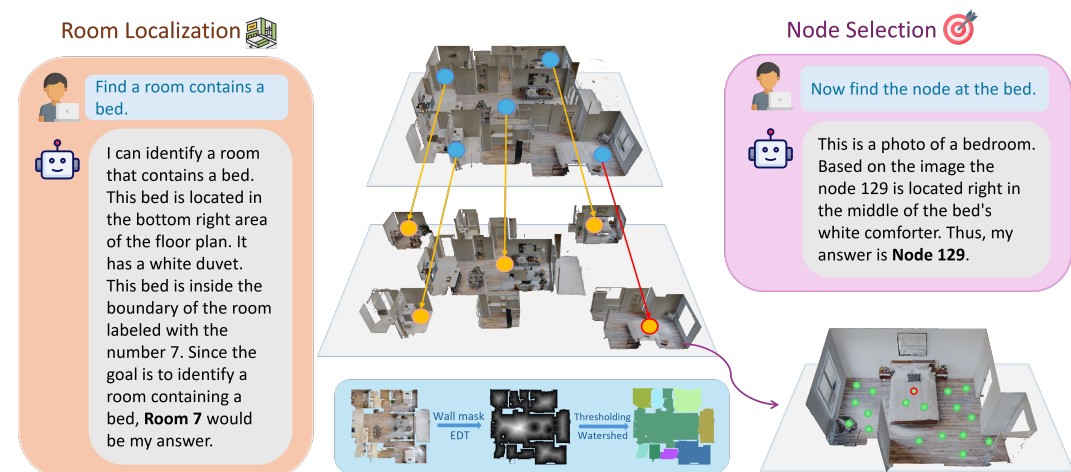

Figure 3: For global reasoning, instead of querying the MLLM for a direct coordinate, we devise a hierarchical, two-stage framework, which effectively leverages MLLM's vision priors.

### 3.1 VERSATILE GOAL NAVIGATION

**Task Definition**. We formulate our goal-oriented navigation task as follows: An embodied agent is deployed in an indoor environment $E$. The agent's objective is to navigate to a goal $g$, which in turn defines a target object instance $O$ or multiple possible instances within object category $c$ in the environment. The goal $g$ can be specified in one of three ways, defining three distinct sub-tasks: Object-goal Navigation, ON (Chaplot et al., 2020), where $g$ is an object category, Instance-Image-goal Navigation, IIN (Krantz et al., 2022), where $g$ is an image containing object that can be found in the scene and Text-goal Navigation, TN (Sun et al., 2025), where $g$ is a description about a certain object).

The agent is initialized at an arbitrary starting pose $p_0 \in E$. At each timestep $t$, the agent receives an egocentric RGB-D observation $o_t$. It also has access to a global 2D map $M$ and its own pose $p_t$ localized within this map. Based on these inputs, the agent must select an action $a_t$ from a discrete action space $\mathcal{A} = \{\text{move\_forward}, \text{turn\_left}, \text{turn\_right}, \text{stop}\}$. The task is successfully done if the agent stops within $d$ meters of $O$ in less than $T$ time steps.

**Task Specification**. Unlike traditional goal-oriented navigation, which requires extensive exploration in the unseen environments, we consider zero-shot object navigation with global information: the task necessitates reasoning on the global view, the goal $g$ can be freely specified with text (a category or a description) or a image and the navigation system works in a training-free manner.

### 3.2 GLOBAL REASONING

The global reasoning module aims to translate the goal description into a specific, long-term navigational goal on the 2D map, denoted as a coordinate $p_{\text{global}}$. To achieve this, we leverage the inherent commonsense and spatial reasoning capabilities of a Multi-modal Large Language Model (MLLM). Instead of directly regressing coordinates, which is often imprecise for MLLMs, we propose a hierarchical, coarse-to-fine framework. As illustrated in Figure 3, this two-stage approach first localizes the target to a specific room and then pinpoints a location within it, efficiently pruning the search space and simplifying the reasoning task.

**Room-Level Localization**. The first stage narrows the search down to a single room or a distinct region. We begin by processing the global 2D map to create a structured representation suitable for the MLLM. First, a binary wall mask $M_0$ is extracted using thresholding. For room segmentation, we apply Euclidean Distance Transform on $M_0$ and derive a number of isolated region seeds by thresholding Euclidean Distance Field (EDF). Then we apply the Watershed algorithm on these region seeds to obtain 2D region masks. This process partitions the entire floor plan into a set of unlabeled, segmented regions. To avoid that the rooms are incorrectly over-segmented, we then merge the tiny regions and derive the final region set $R = \{r_1, r_2, \ldots, r_k\}$:

$$R = \text{Merge}\left(\text{Watershed}(\text{EDT}(M_0, d_{th}), M_0)\right)$$

where $d_{th}$ is the distance threshold to achieve the region seeds on the EDF. Please see B for details.

We then generate an annotated map $M(R)$ labeling each region. The MLLM is prompted with this map and a query $P_1(g)$ to select the most probable room $r^*$. For ON and TN, the prompt is purely text. For IIN, the prompt also contains a goal image of $O$. The MLLM analyzes the spatial layout to identify the target room:

$$r^* = \text{MLLM}(M(R), P_1(g)) = \arg\max_{r_j \in R} P(g \text{ in } r_j \mid M)$$

where $P(g \text{ in } r_j \mid M)$ represents the likelihood of finding the target object $g$ in room $r_j$, given the map $M$. The output of this stage is the selected room $r^*$.

**Intra-Room Node Selection**. To avoid the difficulty of regressing continuous coordinates, we discretize the entire navigable area of the global map into a set of candidate points. Leveraging the wall mask $M_0$ obtained from map preprocessing, we employ Poisson Disk Sampling (PDS) with a sampling radius of $d_s$ on all navigable areas (i.e., regions not covered by $M_0$). PDS ensures that the sampled nodes are uniformly distributed and maintain a minimum distance from each other, providing good coverage and density across the entire map. This process generates a global set of candidate nodes, $N_{\text{global}}$.

Upon selecting a target room $r^*$, we then filter $N_{global}$ to obtain a subset of candidate nodes $N = \{n_1, n_2, \ldots, n_m\}$ that are located strictly within $r^*$. We then generate an annotated map, showing the cropped view of room $r^*$ with these nodes marked with unique numbers. The MLLM is then prompted with the original cropped map $M_{r^*}$ of room $r^*$, the node-annotated room map $M_{r^*}(N)$, along with the designed prompt $P_2(g)$ to the goal $g$, selecting the most plausible node $n^* \in N$:

$$n^* = \text{MLLM}(M_{r^*}(N), M_{r^*}, P_2(g)) = \arg\max_{n_j \in N} P(g \text{ at } n_j \mid M_{r^*})$$

where $M_{r^*}$ is the map of the selected room. The 2D coordinates of this chosen node $n^*$ become the final global goal, $p_{\text{global}}$, which is then passed to the local navigation method.

## 3.3 Local Navigation and Target Verification

**Local Navigation**. The local navigation module guides the agent safely and efficiently to the global target $p_{\text{global}}$. Unlike conventional navigation, $p_{\text{global}}$ often lies in unexplored regions, meaning a complete, direct path may not exist in the agent's current map. Our approach employs a hierarchical control strategy to dynamically plan collision-free and optimal paths.

Central to the navigator is an online occupancy map, acting as the agent's long-term memory. This map categorizes areas into explored, unexplored, and occupied. Initially, occupied areas are defined by the wall mask $M_0$, with continuous updates from RGB-D observations during navigation.

To navigate towards $p_{\text{global}}$, A* search is executed every $T$ timesteps on the latest occupancy map to find an optimal path. A short-term waypoint $w_t$, $d_0$ meters ahead along this path, then serves as the immediate goal for the low-level controller. EDF, derived from EDT, acts as an additional costmap during A* search to ensure waypoints avoid walls and obstacles.

For reactive, collision-free movement towards $w_t$, the Vector Field Histogram* (VFH*) algorithm (Ulrich & Borenstein, 2000) computes steering commands by analyzing local obstacles in the occupancy map. A safety check ensures robustness: if online map updates reveal $w_t$ or $p_{\text{global}}$ is within an occupied area, the point is immediately relocated to the nearest valid, non-occupied location, preventing the agent from getting stuck. The agent iteratively follows these waypoints. Local navigation terminates and transitions to the target verification phase when the agent reaches a predefined proximity to $p_{\text{global}}$.

**Target Verification**. Upon reaching this first proximity threshold, the agent transitions into a verification mode to confirm the presence of the target object $O$. At every time step, the agent attempts to detect the target object within its current field of view using an object detector.

If no confident detection for the target category $c$ is made in this initial view, the agent performs a final short approach to a second, closer proximity threshold, using VFH* with the original $p_{\text{global}}$ as its waypoint. If still no detection at this second threshold, the agent then performs a 360-degree in-place scan. At each rotational step, the egocentric RGB image is processed by the object detector.

If a confident detection for the target category $c$ is made at any stage of the verification process, we proceed to precise 3D localization. The 2D bounding box is fed to MobileSAM for precise segmentation mask generation. This mask isolates corresponding depth image points, which are then back-projected using camera intrinsics to form a 3D point cloud of the target. Its centroid in the occupancy map is computed to determine the object's exact position. The agent then navigates to this precise position using VFH* with the detected object's centroid as its waypoint, and executes a 'stop' action, completing the task. If the agent completes the 360-degree scan without any confident detections, the 'stop' action is called as well.

### 3.4 MODEL ENSEMBLE

To further enhance performance in global reasoning, we design an additional plug-and-play component that integrates the strengths of different models. For our global reasoning task "locating objects from a top-down map", it typically lack large-scale datasets for MLLM training. We recognize that different MLLMs exhibit varying capabilities and limitations across different scenarios. This implies that simply switching to another model might not always significantly enhance performance. Therefore, we propose a model ensemble approach, introducing an additional discriminator to select the more plausible $p_{\text{global}}$ from those provided by Model A and Model B. This aims to strengthen the performance of our global reasoning component.

Specifically, we employ two independent Global Reasoning (GR) units, denoted as $\text{GR}_A$ and $\text{GR}_B$. These two units are based on different MLLM A and B. Each reasoning unit independently receives the global 2D map $M$ and the goal $g$, and performs room-level localization and intra-room node selection to generate a candidate global target point:

$$p_{\text{global}}^A = \text{GR}_A(M, g)$$

$$p_{\text{global}}^B = \text{GR}_B(M, g)$$

To enable the discriminator to compare these two candidate points, we visualize them on the map. We generate two cropped annotated maps: $M_A$ and $M_B$, where $p_{\text{global}}^A$ and $p_{\text{global}}^B$ are marked on two cropped map $M_a$ and $M_b$ from $M$ respectively.

Subsequently, these two annotated maps, along with the original goal $g$, are fed into an additional discriminator MLLM. The task of this discriminator MLLM is to evaluate which candidate point better aligns with the semantics of goal $g$, based on its understanding of the map layout and the goal description. The discriminator is guided by a carefully designed prompt $P_{\text{dis}}$ that encourages the MLLM to perform "self-verification," comparing the plausibility of the two proposals,

$$p_{\text{global}}^{\text{final}} = \text{Discriminator}(M_A, M_B, P_{\text{dis}}(g))$$

The discriminator MLLM then outputs the global target point $p_{\text{global}}^{\text{final}}$ that it deems most plausible. This finally selected $p_{\text{global}}^{\text{final}}$ is then passed to the local navigation module, serving as the ultimate target for the agent's navigation. This ensemble approach leverages the complementary strengths of multiple MLLMs and enhances the robustness and accuracy of global reasoning through the discriminator's verification mechanism.

## 4 EXPERIMENTS

We validate our framework through extensive experiments, comparing it with state-of-the-art methods across multiple benchmarks. Ablation studies confirm the effectiveness of our core components, and qualitative results demonstrate its generalization to complex scenarios like multi-floor and multi-agent navigation. Readers are encouraged to view our supplementary video.

### 4.1 EXPERIMENT SETUP

**Benchmarks**. We evaluate our ReasonNav framework on object-goal, image-goal and text-goal navigation. For object-goal, we conduct experiments on the widely used Habitat-Matterport 3D (HM3D) 2022 challenge benchmark. For image-goal and text-goal navigation, we compare with other methods on HM3D 2023 ImageNav challenge and TextNav following Sun et al. (2025).

Table 1: Results of Obj-goal, Image-goal and Text-goal navigation on HM3D challenge benchmarks. We compare the SR and SPL of state-of-the-art methods in different settings.

| Method | Training -Free | ObjNav | | ImgNav | | TextNav | |
|---|---|---|---|---|---|---|---|
| | | SR | SPL | SR | SPL | SR | SPL |
| OVRL-v2 (Yadav et al., 2023) | ✗ | 64.7 | 28.1 | – | – | – | – |
| OVRL-v2-IIN (Yadav et al., 2023) | ✗ | – | – | 24.8 | 11.8 | – | – |
| IEVE (Lei et al., 2024) | ✗ | – | – | 70.2 | 25.2 | – | – |
| PSL (Sun et al., 2025) | ✗ | 42.4 | 19.2 | 23.0 | 11.4 | 16.5 | 7.5 |
| GOAT (Chang et al., 2023) | ✗ | 50.6 | 24.1 | 37.4 | 16.1 | 17.0 | 8.8 |
| ESC (Zhou et al., 2023b) | ✓ | 39.2 | 22.3 | – | – | – | – |
| OpenFMNav (Kuang et al., 2024) | ✓ | 54.9 | 24.4 | – | – | – | – |
| Mod-IIN (Krantz et al., 2023) | ✓ | – | – | 56.1 | 23.3 | – | – |
| SG-Nav (Yin et al., 2024) | ✓ | 54.0 | 24.9 | – | – | – | – |
| VLFM (Yokoyama et al., 2023) | ✓ | 52.5 | 30.4 | – | – | – | – |
| Trihelper (Zhang et al., 2024b) | ✓ | 56.5 | 25.3 | – | – | – | – |
| InstructNav (Long et al., 2024) | ✓ | **58.0** | 20.9 | – | – | – | – |
| UniGoal (Yin et al., 2025) | ✓ | 54.5 | 25.1 | **60.2** | 23.7 | 20.2 | 11.4 |
| **ReasonNav (ours)** | ✓ | 57.9 | **31.4** | 47.8 | **30.4** | **38.8** | **24.3** |

**Implementation Details**. Our evaluation was conducted in Habitat-sim. We set the maximum time steps $T = 500$, with a linear step size of 0.25m and an angular rotation of 30 degrees per action. We also set the success threshold $d = 1$m. In our PDS sampleing process, we use a sampling radius of 0.5m. We used Seed-1.6-thinking and Gemini-2.5 pro as global reasoning model, while utilizing GPT-5 as discriminator. Detailed prompts we are using are provided in Appendix A.

**Evaluation Metrics**. We use two standard metrics: Success Rate (SR), the percentage of successful episodes, and Success weighted by Path Length (SPL), which measures path efficiency. If an episode is successful, SPL = $\frac{\text{Optimal Path Length}}{\text{Path Length}}$ , otherwise SPL = 0.

## 4.2 COMPARISION WITH STATE-OF-THE-ART

We compare ReasonNav with the state-of-the-art goal-oriented navigation methods of different settings across three tasks in Table 1.

**Object-goal Navigation**. ReasonNav achieves the highest SPL among all methods, including trained ones, underscoring its superior path efficiency. Despite a marginally lower SR than Instruct-Nav (Long et al., 2024), our significantly higher SPL (31.4% vs. 20.9%) highlights our ability to find more direct paths to the goal. Our method also surpasses many other training-free and fine-tuned approaches in overall performance.

**Image-goal Navigation**. ReasonNav adopts a unified framework for all navigation tasks, which means it relies on object detectors for final target verification in ImgNav, rather than specialized similarity matching techniques. This design choice, while ensuring broad applicability, leads to an SR (47.8%) that is slightly lower than some highly specialized methods. However, ReasonNav still achieves the highest SPL (30.4%) in this category. This high SPL stems directly from our global reasoning, which pinpoints a precise waypoint and eliminates the need for the extensive local exploration and matching required by other methods.

**Text-goal Navigation**. The inherent strength of ReasonNav's MLLM-powered global reasoning is particularly evident in TextNav. Here, our framework demonstrates clear dominance, achieving the best performance across all metrics with an SR of 38.8% and an SPL of 24.3%. This significantly surpasses other methods like GOAT (Chang et al., 2023) and UniGoal (Yin et al., 2025), highlighting ReasonNav's superior ability to interpret complex textual instructions and translate them into precise navigation goals in a zero-shot manner.

In summary, ReasonNav's strength lies in its human-like global reasoning, which translates diverse goals into precise map coordinates. This training-free, unified approach consistently yields high SPL by enabling direct, efficient paths to the target, setting a new benchmark for zero-shot navigation, especially in text-goal tasks where its semantic understanding excels.

Table 2: Ablation study on the selection module. We compare our proposed multi-stage selection against directing predicting coordinate, and a single-stage baseline on the HM3D ObjNav challenge dataset. We are using Seed-1.6-Thinking as reasoning model in this experiment.

| Selection Method | SR | SPL |
| --- | --- | --- |
| Directly predicting coordinate | 12.3 | 6.13 |
| Single-stage selection | 44.5 | 23.1 |
| Multi-stage selection | **55.1** | **29.6** |

Table 3: Ablation study on different reasoning MLLMs. We report the performance of our framework when equipped with different large language models on the HM3D ObjNav, ImgNav, and TextNav datasets.

| Reasoning MLLM | ObjNav | | ImgNav | | TextNav | |
| --- | --- | --- | --- | --- | --- | --- |
| | SR | SPL | SR | SPL | SR | SPL |
| Qwen2.5-7B-VL | 41.3 | 21.2 | 22.8 | 14.3 | 17.0 | 10.1 |
| Seed-1.6-Thinking | 55.1 | 29.6 | 40.2 | 25.2 | 30.1 | 19.0 |
| Gemini-2.5-Pro | 55.8 | 29.1 | 40.7 | 25.6 | 37.2 | 22.8 |
| Model Ensemble | **57.9** | **31.4** | **47.8** | **30.4** | **38.8** | **24.3** |

## 4.3 ABLATION STUDY

In our ablation study, we validate the effectiveness of our proposed multi-stage selection module and conduct experiments to compare the performance of different reasoning LLMs.

**Ablation on our multi-stage selection**. Our ablation study, presented in Table 2, clearly demonstrates the efficacy of our multi-stage selection strategy. Directly predicting continuous 3D coordinates yielded extremely low performance (SR 12.3%, SPL 6.13%), indicating the MLLM's difficulty in regressing precise spatial coordinates and underscoring the necessity of selecting discrete, navigable nodes. While single-stage selection, which involves the MLLM choosing from the entire global topological graph, showed improved results (SR 44.5%, SPL 23.5%). However, it still shows asking the MLLM to select the optimal node from a vast global graph in a single step remains a challenging task. In contrast, our proposed multi-stage selection method achieved the best performance (SR 55.1%, SPL 29.6%), significantly outperforming single-stage selection (SR by 10.6%, SPL by 6.1%). This ablation study thus clearly confirms the superiority of ReasonNav's multi-stage node selection strategy, demonstrating that its hierarchical decision-making, which breaks down the complex target selection task into more manageable steps, enables the MLLM to reason and select targets with greater precision and robustness.

**Ablation on different reasoning MLLMs.** To evaluate the impact of the reasoning engine on our framework, we benchmarked several MLLMs, including the open-source Qwen2.5-7B-VL, and the advanced proprietary models Seed-1.6-Thinking and Gemini-2.5-Pro. The results, detailed in Table 3, show a clear correlation between the model's reasoning capabilities and navigation performance. The advanced reasoning models, Seed-1.6-Thinking and Gemini-2.5-Pro, achieves significantly better performance compared with the smaller model across all the tasks, highlighting the importance of advanced reasoning for complex spatial understanding for our framework. Gemini-2.5-pro and Seed-1.6-Thinking generally derive similar results in object-goal navigation and image-goal navigation. However, possibly due to stronger capability in text understanding and reasoning, Gemini-2.5-pro outperforms Seed-1.6-Thinking with a large margin on TextNav. To maximize performance, we implement the model ensemble strategy in our method. This approach yielded the best results across all tasks. This demonstrates that ensembling leverages the diverse strengths of each model, mitigating individual weaknesses and leading to more robust and accurate navigation decisions.

This analysis confirms that advanced MLLMs are critical for performance. Our model ensemble strategy, by leveraging the complementary strengths of different models, consistently achieves the best results and pushes our framework to state-of-the-art performance.

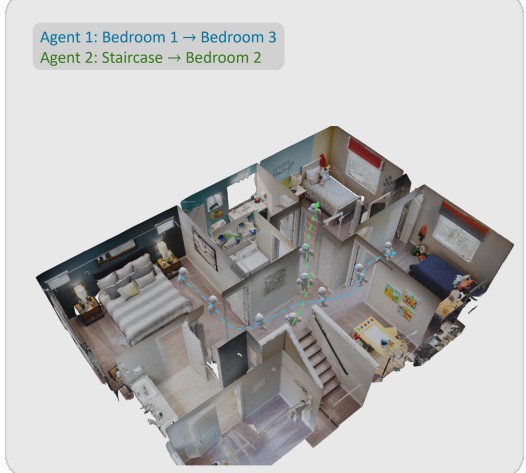

(a) Figure 4a: Our framework can tackle the multi-floor task by decomposing it to single-floor tasks.

(b) Figure 4b: Our framework inherently allows different agents running in the same workspace.

## 4.4 QUALITATIVE ANALYSIS

### 4.4.1 NAVIGATION IN MULTI-FLOOR SCENARIOS

Despite the inherent challenges of precisely controlling an agent to traverse between different floors via stairs in a zero-shot scenario, our proposed method exhibits powerful inter-floor reasoning abilities. As indicated in Figure 4a, when a target object is localized on a distinct floor, we utilize Chain-of-Thought (CoT) to decompose the complex multi-floor path into a series of single-floor navigation tasks, specifically, moving from the current position to the stairs, and then from the stairs to the target. This represents a crucial step beyond traditional single-floor navigation paradigms, enabling agents to tackle more realistic and complex spatial reasoning tasks.

### 4.4.2 NAVIGATION WITH MULTIPLE AGENTS

In real-world scenarios, the presence of dynamic obstacles (e.g., moving objects) poses significant challenges to dense scene modeling approaches reliant on point cloud representations. This complexity leads to critical issues in multi-agent collaborative navigation, where concurrent operations within shared environments induce catastrophic interference in existing frameworks. Prior construction-based methods often rely on static or one-shot semantic information. The dynamic presence of additional agents can substantially interfere with their semantic understanding, leading to compromised navigation performance. As shown in Figure 4b, our approach, on the other hand, fundamentally differs by relying solely on depth data for local obstacle avoidance during navigation. This design choice inherently minimizes inter-agent conflicts, thereby demonstrating superior scalability and robustness in multi-agent environments.

## 5 CONCLUSION

In this paper, we introduce ReasonNav, a novel, human-inspired framework for zero-shot embodied navigation. Our approach uniquely leverages the strengths of MLLMs for high-level hierarchical global reasoning, while relegating low-level control and execution to robust, deterministic planners. This "reason-then-act" paradigm sidesteps the challenges of using LLMs for precise, continuous control, a task for which they are not optimized. We demonstrate that only by focusing the MLLM's role on a one-off global reasoning task with one global view image, we can achieve state-of-the-art (SOTA) performance in a computationally efficient manner. The framework's zero-shot and computationally efficient nature eliminates the need for costly and time-consuming fine-tuning, RL training or real-time inference, making it a scalable and practical solution for a diverse range of embodied navigation tasks.

## 6 ETHICS STATEMENT

All research presented in this paper was conducted in adherence to the ICLR Code of Ethics. Our work is centered on a framework for embodied navigation within simulated environments. The experiments exclusively utilize the publicly available HM3D dataset, which consists of static 3D reconstructions of indoor spaces and does not involve human subjects, personally identifiable information, or sensitive data. The goal of this research is to advance the state of the art in embodied navigation, and we do not foresee any direct negative societal impacts stemming from this work.

## 7 REPRODUCIBILITY STATEMENT

We have made a concerted effort to ensure the reproducibility of our work. Our framework, Reason-Nav, is described in detail in Section 3, with its core components—global reasoning and local navigation—clearly outlined. To facilitate replication, the Appendix B provides comprehensive pseudocode for our key algorithms, including Room Segmentation, Node Generation, and the complete Local Navigation and Target Verification pipeline. The experimental setup, including the specific benchmarks, evaluation metrics, and implementation details is specified in Section 4.1. To ensure the central reasoning component of our method is reproducible, we have included the exact, detailed prompts used to query the MLLMs for all stages of our framework in Appendix A.

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

## APPENDIX

In this appendix, we present:

- Prompts used in ReasonNav.
- Details of our method.
- Additional comparison with concurrent work.
- Limitations and future work.
- The use of large language models (LLMs)

## A  PROMPTS USED IN REASONNAV

We provide our detailed prompts used in ReasonNav:
**Room-Level Localization:**

```
You are an AI assistant for a robot's navigation system.  Your
task is to identify a certain room.
CONTEXT: You will be provided with a top-down floor plan
showing furniture and layout with room segmentation.  (The
floor plan is segmented to at least one room/region with
numbered room annotations.  The boundaries of different rooms
is noted in white.)
GOAL: Identify the room number that contains the {goal_object}.
INSTRUCTIONS:
1.  If you can find the object directly from map, describe the
object's location using text.
2.  Analyze the Goal Object:  Determine the room number
where the {goal_object} can be found.  If you cannot find
{goal_object} on the top-down view, then please consider
the most-possible room that the {goal_object} may locate at.
(e.g., a "bed" is in a bedroom)
3.  Verify the Room:  Scan the top-down view to make sure that
the
{goal_object} indeed locates in the chosen room.  If
{goal_object} cannot be found, make sure the chosen room
is the most appropriate room to search.  Note that you
should consider the precise semantics of {goal_object} during
verification.  (e.g., "sofa chair" is different from "sofa")
Note that you should choose the room that contains the
{goal_object} instead of the room that is the closest to the
{goal_object}!
Repeat the instruction until the verification (3.)  is passed.
Provide your answer in the last line in the form of:  "Room X"
where X is your chosen number.  (e.g.  Room 1)
Goal Object:  {goal_object}
```

where {goal_object} will be replaced by the goal $g$.

**Intra-Room Node Selection:**

```
CONTEXT: You are given two images:
1.  Map Image:  A top-down schematic of the room's layout.
2.  Node Image:  The same map with numbered navigation nodes.
GOAL: {goal_object}
INSTRUCTIONS:
1.  Analyze the Goal:  Consider the common placement of a
{goal_object}.  For example, a "TV" is opposite to sofa in
the living room; a "book" is on a shelf or table.  Note the
precise semantics of {goal_object} and do not misunderstand the
target.  (e.g., sofa chair is different from sofa).
2.  Locate on Map:  Scan the Map Image to find the
{goal_object} or the most logical place it would be (e.g., find
the dining table if the goal is a "plate").
3.  Select Best Node:  Based on your location analysis, choose
the single best node from the Node Image.  The best node is
determined by this priority:
*  Priority 1:  A node located directly on the object.
*  Priority 2:  If no node is on the object, the node closest
to the object.
*  Priority 3:  If the object cannot be found on the
topdown-view, the node that provides the best vantage point
to search the inferred area.
4.  Verify The Node:  Make sure the selected node satisfy the
above requirements.
Repeat the instructions until the verification (4.)  is
passed.
Goal Object:  {goal_object} Provide your answer in the last
line in the form of:  "node X" where X is your chosen number.
(e.g.  node 100)
```

where {goal_object} will be replaced by the goal *g*.

**Discriminator:**

```
You are an expert navigation system evaluator.  I need you to
analyze a controversial episode where two different AI models
disagreed on the success of an object navigation task.
Target Object: {goal_object}
Images Provided:  You will see two separate images:
1. First Image (Model 1):  Shows an area around Model 1's
target selection with a BLUE circle marking the chosen target
location.
2. Second Image (Model 2):  Shows a area around Model 2's
target selection with a RED circle marking the chosen target
location.
Your Task:  Please analyze these navigation scenarios and
determine which model made the better decision.  Consider:
1. Target Identification:  Which model identified a more
plausible target location for "{goal_object}"?  Look at the
environment around each colored circle, find the corresponding
node that is closer to the {goal_object}.
2. Accessibility:  If two nodes are both at a plausible
position of the {goal_object}, which target location appears
more accessible and reachable in a real navigation scenario?
(i.e.  more close to the navigable/walkable area and more
close to the open space. (e.g.  Two chairs.  The one that
is closer to the open area is more suitable than the one that
is closer to the wall.))
Please respond with:
1.  Your analysis of both models' target selections based on
the environmental context
2.  Which model you believe made the better decision (Model 1
or Model 2)
3.  Key reasoning points for your decision
Output Format:  Decision:  [Model 1 or Model 2]
```

where {goal_object} will be replaced by the goal $g$.

## B DETAILS OF OUR METHOD

### B.1 ROOM SEGMENTATION

Our room segmentation pipeline is designed to robustly partition a top-down floor plan into distinct room areas. The process leverages a combination of morphological operations, EDT, and the Watershed algorithm, followed by a post-processing step to merge small, erroneous room fragments.

The process begins with a binary wall mask, where non-walkable areas (walls) are distinguished from walkable spaces. A morphological closing operation is first applied to fill minor gaps and holes in the detected walls, ensuring that rooms are well-enclosed entities.

Subsequently, the EDT is computed on the walkable areas. The EDT assigns each pixel a value corresponding to its distance to the nearest wall. This transform is crucial as it allows us to identify the central, core regions of rooms—pixels with high distance values are far from any walls and are thus strong candidates for being region seeds.

The determination of these seeds is critical. For greater adaptability, we employ Otsu's method on the normalized and blurred distance map. This automatically finds an optimal threshold to separate the map into two classes: room cores (sure foreground) and areas closer to walls (ambiguous regions). This dynamic approach makes the segmentation robust to varying room sizes and layouts.

With the seeds (sure foreground) and walls (sure background) identified, the Watershed algorithm is applied. This algorithm treats the distance map as a topographical surface, "flooding" it from the seed locations until the "waters" from different seeds meet. These meeting lines form the boundaries that segment the space into initial room candidates.

Finally, a merging procedure is executed to handle over-segmentation, where small, insignificant regions might be incorrectly labeled as separate rooms. This iterative process identifies rooms smaller than a minimum area threshold and merges them into the most suitable adjacent larger room. The suitability is determined by a merge score that considers the length of the shared boundary and the proximity of the room centroids, ensuring that merges are geometrically logical. The process is outlined in Algorithm 1.

---

**Algorithm 1** Room Segmentation via EDT and Watershed

---

**Require:** Binary Wall Mask $M_{\text{binary}}$
**Ensure:** Labeled Marker Matrix $M_{\text{rooms}}$
$\quad M_{\text{closed}} \leftarrow \text{MorphologicalClose}(M_{\text{binary}})$
$\quad M_{\text{dist}} \leftarrow \text{EuclideanDistanceTransform}(M_{\text{closed}})$
$\quad M_{\text{norm}} \leftarrow \text{Normalize}(M_{\text{dist}}, \text{range} = [0, 255])$
$\quad M_{\text{blur}} \leftarrow \text{GaussianBlur}(M_{\text{norm}})$
$\quad T_{\text{Otsu}}, M_{\text{seeds}} \leftarrow \text{OtsuThreshold}(M_{\text{blur}})$
$\quad M_{\text{bg}} \leftarrow \text{Dilate}(M_{\text{closed}}, K)$
$\quad M_{\text{unknown}} \leftarrow M_{\text{bg}} - M_{\text{seeds}}$
$\quad M_{\text{markers}} \leftarrow \text{ConnectedComponents}(M_{\text{seeds}})$
$\quad M_{\text{markers}} \leftarrow M_{\text{markers}} + 1$
$\quad M_{\text{markers}}[M_{\text{unknown}} = \text{true}] \leftarrow 0$
$\quad M_{\text{rooms}} \leftarrow \text{Watershed}(M_{\text{closed}}, M_{\text{markers}})$
$\quad M_{\text{rooms}} \leftarrow \text{MergeSmallRooms}(M_{\text{rooms}})$
$\quad$ **return** $M_{\text{rooms}}$

---

### B.2 NODES GENERATION

Our implementation begins with the binary walkable mask and applies a configurable wall padding. This is achieved through morphological erosion, which shrinks the walkable area by a specified distance. This crucial preprocessing step ensures that all generated navigation nodes maintain a safe distance from walls and obstacles, preventing the planner from creating paths that are too close to collisions.

Then we use an enhanced PDS algorithm to handle environments with multiple disconnected regions. First, a connected components analysis is performed on the padded walkable mask to identify all distinct, navigable areas. Then, the PDS algorithm is executed independently within each of these regions. This ensures that every separate walkable space is populated with navigation nodes, guaranteeing comprehensive coverage of the entire environment. The algorithm iteratively adds points by selecting a random point from an "active list," generating new candidate points in an annulus around it, and validating that the new point respects the minimum distance constraint relative to all existing points. The process is outlined in Algorithm 2.

### B.3 LOCAL NAVIGATION

The entire process is outlined in Algorithm 3.

---

**Algorithm 2** Node Generation via Multi-Region Poisson Disk Sampling

---

**Require:** Binary Walkable Mask $M_{\text{walkable}}$, Wall Padding Distance $D_{\text{padding}}$, PDS Radius $R_{\text{pds}}$
**Ensure:** Set of Navigation Nodes $N_{\text{all}}$

    $M_{\text{padded}} \leftarrow \text{Erode}(M_{\text{walkable}}, D_{\text{padding}})$
    $C_{\text{list}} \leftarrow \text{FindConnectedComponents}(M_{\text{padded}})$
    $N_{\text{all}} \leftarrow \emptyset$
    **for** each component $C$ in $C_{\text{list}}$ **do**
        $N_{\text{active}} \leftarrow \emptyset$
        $N_{\text{component}} \leftarrow \emptyset$
        $G_{\text{grid}} \leftarrow \text{InitializeGrid}(C, R_{\text{pds}})$
        $p_{\text{initial}} \leftarrow \text{SelectRandomPointIn}(C)$
        Add $p_{\text{initial}}$ to $N_{\text{active}}$, $N_{\text{component}}$, and $G_{\text{grid}}$
        **while** $N_{\text{active}}$ is not empty **do**
            $p_{\text{current}} \leftarrow \text{SelectRandomPointFrom}(N_{\text{active}})$
            $p_{\text{found}} \leftarrow \text{false}$
            **for** $i \leftarrow 1$ to $k_{\text{attempts}}$ **do**
                $p_{\text{new}} \leftarrow \text{GeneratePointInAnnulus}(p_{\text{current}}, R_{\text{pds}}, 2R_{\text{pds}})$
                **if** $\text{IsValid}(p_{\text{new}}, C, G_{\text{grid}}, R_{\text{pds}})$ **then**
                    Add $p_{\text{new}}$ to $N_{\text{active}}$, $N_{\text{component}}$, and $G_{\text{grid}}$
                    $p_{\text{found}} \leftarrow \text{true}$
                    **break**
                **end if**
            **end for**
            **if** not $p_{\text{found}}$ **then**
                Remove $p_{\text{current}}$ from $N_{\text{active}}$
            **end if**
        **end while**
        $N_{\text{all}} \leftarrow N_{\text{all}} \cup N_{\text{component}}$
    **end for**
    **return** $N_{\text{all}}$

---

**Algorithm 3** Local Navigation and Target Verification

---

**Require:** Global goal $p_{\text{global}}$, Proximity thresholds $d_{\text{prox1}}, d_{\text{prox2}}$
**Ensure:** Navigation Stop

    $M_{\text{occ}} \leftarrow \text{InitializeMap}()$
    **while** $\text{distance}(\text{GetAgentPose}(), p_{\text{global}}) > d_{\text{prox1}}$ **do**
        $o_{\text{t}} \leftarrow \text{GetCurrentObservation}()$
        $M_{\text{occ}} \leftarrow \text{UpdateMap}(M_{\text{occ}}, o_{\text{t}})$
        $P \leftarrow \text{A\_star\_Search}(M_{\text{occ}}, \text{GetAgentPose}(), p_{\text{global}})$
        $w_{\text{t}} \leftarrow \text{GetLocalWaypoint}(P)$
        $w_{\text{t}} \leftarrow \text{SafetyCheck}(w_{\text{t}}, M_{\text{occ}})$
        $action \leftarrow \text{VFH\_star\_Controller}(w_{\text{t}}, M_{\text{occ}})$
        $\text{Execute}(action)$
    **end while**
    **while** $\text{distance}(\text{GetAgentPose}(), p_{\text{global}}) > d_{\text{prox2}}$ **do**
        $found, p_{\text{obj}} \leftarrow \text{Scan360AndDetect}()$
        **if** $found$ **then**
            **break**
        **end if**
        $\text{ApproachCloser}(p_{\text{global}})$
    **end while**
    **if** not $found$ **then**
        $found, p_{\text{obj}} \leftarrow \text{Scan360AndDetect}()$
    **end if**
    **if** $found$ **then**
        $p_{\text{final}} \leftarrow \text{LocalizeObject3D}(p_{\text{obj}})$
        $\text{NavigateToFinalPosition}(p_{\text{final}})$
    **end if**
    Navigation Stop

---

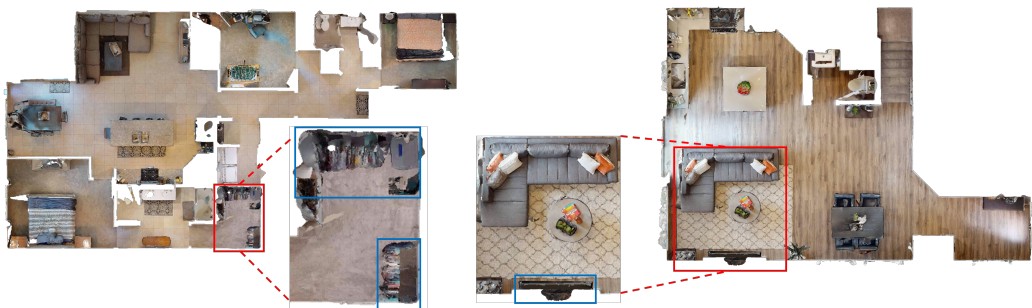

Figure 5: The ambiguity of the global map may influence the outcome. For example, MLLM sometimes struggles to pin-point the clothes (at the right-bottom corner) on the left map or the tv monitor (at the left-bottom corner) on the right map. (denoted with blue bounding box)

## C  ADDITIONAL COMPARISON WITH CONCURRENT WORK

During our research, we noted the concurrent development of WMNav (Nie et al., 2025), another framework that leverages Vision-Language Models (VLMs) for zero-shot object navigation. Both ReasonNav and WMNav achieve state-of-the-art performance on the HM3D ObjectNav benchmark, demonstrating remarkably similar top-line metrics (ReasonNav: 57.9% SR, 31.4% SPL; WMNav: 58.1% SR, 31.2% SPL). Despite these comparable results, the two methods are founded on fundamentally different philosophies regarding the role of the MLLM in navigation.

The core distinction lies in the source of visual understanding and the frequency of MLLM inference. ReasonNav operates on a *global-view*, *one-off* reasoning paradigm. It leverages the MLLM's powerful spatial and semantic understanding capabilities a single time at the beginning of an episode. By processing a complete top-down map, ReasonNav formulates a comprehensive global plan, identifying a precise target coordinate ($p_{\text{global}}$) before the agent begins to move. This "reason-then-act" approach cleanly separates high-level strategic planning from low-level, deterministic execution, minimizing computational overhead and avoiding the risk of cumulative errors from repeated model inferences.

In stark contrast, WMNav relies entirely on *local observations* and employs a *step-by-step* reasoning process. Its visual understanding is derived from panoramic images captured at the agent's current position at each timestep. This approach necessitates a heavy, iterative reliance on the VLM throughout the exploration phase to interpret the immediate surroundings, predict the potential presence of the target in different directions, and update a "Curiosity Value Map". While this allows the agent to react to newly observed information, it involves a substantial number of VLM calls, making the process computationally intensive and potentially susceptible to model hallucinations over long horizons.

In summary, while both frameworks successfully utilize the power of modern foundation models, ReasonNav's approach of using the MLLM for a single, decisive act of global reasoning based on a complete map proves to be a more efficient and robust strategy. Our significantly higher SPL, despite a marginal difference in SR, suggests that our global, one-off planning leads to more direct and efficient paths. This architectural choice not only enhances computational efficiency but also aligns more closely with human-like navigation, where a global plan (e.g., looking at a map) precedes local action.

## D  LIMITATIONS AND FUTURE WORK

### D.1  LIMITATIONS

**Reliance on a high-quality global view:** The current framework fundamentally depends on the availability of a clean, top-down 2D map of the environment. This assumes such prior information is accessible, which may not be the case in many real-world applications.

**Disregarding semantic information from local observations:** ReasonNav performs its MLLM-based reasoning as a one-off step before navigation begins. It does not incorporate semantic information from its egocentric view while in transit to correct or refine its global plan. For example, if the initial plan is incorrect, the agent cannot use local visual cues (e.g., seeing kitchen appliances) to recognize its mistake and re-evaluate its path. Also, it cannot use local observations to refine the reasoning process.

**Inability to reason about objects with ambiguous top-down projections:** The MLLM's ability to locate a target is limited by the quality of the 2D map. Objects that are not clearly or have an indistinct shape from a top-down perspective can be difficult for the MLLM to identify (even though our multi-stage global reasoning consists of "zoom-in" strategy), potentially leading to inaccurate goal localization. For example, as shown in Figure 5, the recognition of the clothes or the tv sometimes is chllenging for MLLMs.

## D.2 FUTURE WORK

**Incorporating diverse global map modalities:** To reduce the reliance on perfect pre-built maps, future work could explore using other forms of global information, such as architectural CAD drawings, hand-drawn sketches, etc.

**Integrating global reasoning with local semantic feedback:** A promising direction is to create a tighter loop between global planning and local perception. The MLLM could be invoked not just at the start but also when the agent encounters local semantic cues. This would allow the agent to re-reason and adapt its strategy mid-journey. Furthermore, this would develop interactive reasoning mechanisms where the agent can perform exploratory actions to resolve uncertainty about a potential target's location, combining the strengths of both global foresight and local observation.

## E THE USE OF LARGE LANGUAGE MODELS (LLMS)

This section discloses the extent to which LLMs were utilized as a general-purpose assistance tool in this research.

The use of LLMs in this work can be categorized into two main areas: academic writing and code generation.

For academic writing, LLMs were employed to enhance the clarity, conciseness, and grammatical accuracy of the manuscript. This included tasks such as rephrasing sentences, improving the flow of arguments, and ensuring consistent terminology throughout the paper.

In terms of code generation, LLMs served as a supplementary tool to aid in the implementation of certain algorithms and components of our proposed framework.

It is important to note that while LLMs provided valuable assistance, the core research ideas, experimental design, and the final interpretation of the results were conceived and executed entirely by the authors.

