# OpenReview forum: "ReasonNav: Human-Inspired Global Map Reasoning for Zero-Shot Embodied Navigation"
_ICLR.cc/2026/Conference — ICLR 2026 Conference Withdrawn Submission_

### Official Review · Reviewer_Zfnt · 2025-10-25

**Soundness:** 3
**Presentation:** 3
**Contribution:** 2
**Rating:** 4
**Confidence:** 4

**Summary:**

The paper proposes ReasonNav, a zero-shot embodied navigation framework that separates high-level reasoning and low-level control. A multimodal LLM performs global reasoning on a top-down floor map through a two-stage discrete selection process (room-level then point-level), producing a global goal for a deterministic planner (A* + VFH*). The system supports ObjectNav, ImageNav, and TextNav within a unified, training-free pipeline. Experiments on HM3D demonstrate competitive success rates and state-of-the-art path efficiency (SPL), with ablations validating design choices.

**Strengths:**

1. The paper presents a clear reasoning–planning decomposition, combining LLM-based semantic reasoning with deterministic path planning.
2. The two-stage discrete selection design effectively improves goal localization and navigation efficiency.
3. Experiments show strong zero-shot performance and competitive SPL on standard benchmarks.
4. The paper is well written and reproducible, with detailed prompts, pseudo-code, and ablations.

**Weaknesses:**

1. The paper’s “unified” claim covers only goal-conditioned navigation (ObjectNav, ImageNav, TextNav) and omits instruction-following VLN tasks that require sequential reasoning. Related work such as SmartWay: Enhanced Waypoint Prediction and Backtracking for Zero-Shot Vision-and-Language Navigation (Shi et al., IROS 2025) should be discussed to clarify the scope.
2. Despite the emphasis on reasoning, the proposed method mainly performs static selection (choosing a room and a point) rather than explicit multi-step or interpretable reasoning. The reasoning process is not analyzed or visualized.
3. The approach relies on numerous heuristic components (map segmentation, candidate sampling, prompt templates, ensemble voting), making it highly engineered and potentially sensitive to environment changes.
4. The assumption of an ideal, pre-known floor map limits the method’s applicability in realistic embodied settings.
5. While technically solid, the contribution is primarily system-oriented and lacks deeper insight into reasoning or learning mechanisms.

**Questions:**

1. How would the system perform under incomplete or noisy maps?
2. Would online re-planning or iterative reasoning improve robustness in complex environments?

---

### Official Review · Reviewer_ecbZ · 2025-10-30

**Soundness:** 2
**Presentation:** 2
**Contribution:** 2
**Rating:** 4
**Confidence:** 4

**Summary:**

The paper presents ReasonNav, a human-inspired embodied navigation framework that integrates MLLMs for high-level global reasoning and deterministic planners for local control. The system transforms navigation into a discrete reasoning problem, enabling zero-shot navigation without reinforcement learning or fine-tuning.

**Strengths:**

- The reason-then-act paradigm is well-motivated and effectively addresses the limitations of both end-to-end RL methods and reactive exploration. The analogy to human map-based reasoning is conceptually reasonable.
- The discrete node-based reasoning with hierarchical MLLM prompting, deterministic local planning, and ensemble verification are technically coherent.
- The framework performs competitively without fine-tuning or task-specific training, highlighting scalability with the evolution of foundation models.

**Weaknesses:**

- The method assumes access to a high-quality top-down map, which is unrealistic for many real-world settings where such maps are unavailable or noisy. The authors acknowledge this limitation but should discuss how the system could integrate online SLAM or partial map construction to relax this assumption.
- The reasoning step is performed once at the beginning. The agent does not revise or refine its global plan based on egocentric semantic feedback during navigation. This limits robustness when the MLLM’s initial reasoning is suboptimal. Future work could incorporate dynamic reasoning triggers.
- While the simulation results are comprehensive, there is no demonstration in real robotic environments. Given the deterministic planning nature and reliance on pretrained vision models, an analysis of sim-to-real transfer or physical robot feasibility would strengthen the paper.

**Questions:**

Is it possible to evaluate the model’s performance on popular VLN simulation benchmarks?

---

### Official Review · Reviewer_PSXv · 2025-11-01

**Soundness:** 2
**Presentation:** 2
**Contribution:** 2
**Rating:** 4
**Confidence:** 3

**Summary:**

This paper presents ReasonNav, a framework for zero-shot embodied navigation that decouples global reasoning from local execution. The key innovation is using VLMs to perform hierarchical reasoning over top-down maps. The framework is evaluated on the HM3D benchmark.

**Strengths:**

- The paper is clearly written and easy to follow.
- The appendix provides detailed prompts, supporting reproducibility.

**Weaknesses:**

- The paper fundamentally requires a top-down 2D map with wall information. The authors acknowledge this limitation in the Appendix but don't adequately address how realistic this assumption is. Perhaps in most real-world cases, this assumption would not hold.
- In Table 1, the results of most baselines on ImgNav and TextNav are missing. Given that ReasonNav is not achieving the best SR on both ObjectNav and ImgNav, more results comparison would give a clearer map of the performance of ReasonNav.

**Questions:**

- Computational costs like token usage should be provided.
- Could authors provide more failure case studies? I am wondering what the most common failure modes are.

---

### Note · Authors · 2025-11-13

I have read and agree with the venue's withdrawal policy on behalf of myself and my co-authors.